# The Effects of Orthoptic Therapy on the Surgical Outcome in Children with Intermittent Exotropia: Randomised Controlled Clinical Trial

**DOI:** 10.3390/jcm12041283

**Published:** 2023-02-06

**Authors:** Meiping Xu, Yiyi Peng, Fuhao Zheng, Huanyun Yu, Jiawei Zhou, Jingwei Zheng, Yuwen Wang, Fang Hou, Xinping Yu

**Affiliations:** 1Eye Hospital and School of Ophthalmology and Optometry, Wenzhou Medical University, Wenzhou 325027, China; 2National Clinical Research Center for Ocular Diseases, Wenzhou 325027, China; 3State Key Laboratory of Ophthalmology, Zhongshan Ophthalmic Center, Sun Yat-sen University, Guangzhou 510060, China

**Keywords:** intermittent exotropia, orthoptic therapy, suboptimal surgical outcome, randomised controlled trial, fusional convergence amplitude

## Abstract

Background: To assess the clinical effectiveness of orthoptic therapy in the postoperative stabilisation and rehabilitation of binocular function in children with intermittent exotropia (IXT) after surgery. Methods: This was a prospective, parallel, randomised controlled trial. A total of 136 IXT patients (aged from 7 to 17 years) who had been successfully corrected at 1 month after surgery were enrolled in this study, and 117 patients (58 controls) completed the 12-month follow-up visit. The primary outcome was established as the proportion of patients with suboptimal surgical outcomes, which were defined as: (1) exodeviation ≥10 prism diopters (PD) at distance or near using the simultaneous prism and cover test (SPCT), or (2) constant esotropia ≥6 PD at distance or near using SPCT, or (3) loss of 2 or more octaves of stereopsis from baseline. The secondary outcomes were the exodeviation at distance and near using the prism and alternate cover test (PACT), stereopsis, fusional exotropia control and convergence amplitude. Results: The cumulative probability of suboptimal surgical outcome by 12 months was 20.5% (14/68) in the orthoptic therapy group and 42.6% (29/68) in the control group. There was a significant difference between these two groups (χ^2^ = 7.402, *p* = 0.007). Improvements in stereopsis, fusional exotropia control and fusional convergence amplitude were found in the orthoptic therapy group. A smaller exodrift was found in the orthoptic therapy group at near fixation (t = 2.26, *p* = 0.025). Conclusions: Early postoperative orthoptic therapy can effectively improve the surgical outcome as well as stereopsis and fusional amplitude.

## 1. Introduction

Intermittent exotropia (IXT) is the most common type of childhood-onset exotropia [1,2], and its incidence in China is approximately 3% [3,4,5]. It is a disorder of ocular alignment characterised by an intermittent outward deviation of one eye, which is manifested more frequently with distance viewing, illness, or fatigue. Children with IXT can suffer from a poor negative psychosocial experience [6,7]. When there is poor fusional control of exodeviation and a decrease in stereopsis, surgery is often recommended. The most effective surgical treatment for IXT remains elusive, and surgical outcomes may be influenced by a variety of preoperative (e.g., age, refractive error, anatomical factors, angle of deviation) and postoperative (e.g., early postoperative angle, follow-up period) effects [8]. Postoperative exodrift is common, and initial over-correction has been suggested in order to achieve a successful long-term outcome [9,10,11]. However, patients with early exophoria after surgery could have a high probability of exotropia recurrence [12,13]. Some studies [14,15,16] have supported the idea of pairing surgery with nonsurgical intervention after surgery, such as patching, over-minus correction, or orthoptic therapy. However, whether nonsurgical intervention is indeed effective in permanently alleviating IXT remains unknown.

Orthoptic therapy aims to reduce suppression, stimulate awareness of diplopia, and improve visual function, such as sensory fusion, fusional convergence, accommodation, saccades and proprioceptive awareness [17,18]. Recent studies have demonstrated that fusional convergence parameters are associated with the control of IXT [19,20]. For instance, Yam et al. observed 117 Chinese children with IXT 3 years and reported that the baseline fusional convergence parameters could act as predictive factors in determining the control of IXT [21].

It has been suggested that the combination of surgical intervention and orthoptic therapy can effectively improve the long-term success rate of IXT recovery and enhance binocular function recovery [16,22]. Figueira et al. [23] reported a success rate of 83.3% in treating IXT when surgery was combined with orthoptic therapy and 36.4% when surgery was performed alone at the 2-year follow-up assessment. Martin et al. [24] indicated that office-based vision therapy combined with home reinforcement significantly improved the control of exodeviation in both operated and unoperated IXT. However, there were several limitations in these studies, such as their retrospective design, uneven sample size, potential selection bias, and unique follow-up time points. Hence, despite these studies, it remains unknown whether orthoptic therapy in postoperative IXT patients is effective.

For these reasons, we conducted a randomised controlled study to assess the clinical effectiveness of orthoptic therapy in the stabilisation of postoperative IXT patients. The main purpose of this study was to examine whether orthoptic therapy could improve the surgical outcome as well as binocular functions of IXT paediatric patients after surgery.

## 2. Materials and Methods

### 2.1. Study Design

The design was explained in detail in our previous published protocol [25]. In brief, our study was a prospective, parallel, simple randomised controlled clinical trial. The 2010 consort statement was implemented while reporting the study design, analysis, and interpretations of the results. All procedures met the tenets of the Declaration of Helsinki and were approved by the medical ethics committee of the Affiliated Eye Hospital of Wenzhou Medical University (2019-108-K-101). Written consent and verbal assent were obtained from parents (or guardians) and patients, respectively. The study protocol (V2/2019.08.10) was registered in the Chinese Clinical Trial Registry (ChiCTR1900026891).

### 2.2. Participants’ Inclusion and Exclusion Criteria for the Study

The study was performed within the Affiliated Eye Hospital of Wenzhou Medical University. All participants were postoperative patients with IXT. The decision for undergoing surgery was determined by the surgeon, patient and patient’s parents. The criteria for surgery mainly includes a progressive increase in the angle of exotropia (exo-deviation ≥20 PD at near and distance), exotropia greater than or equal to 50% of waking hours [26,27] or weaker control of exodeviation (office control score ≥3 at near or distance) [28], evidence of progressive loss of stereoacuity, or the appearance of the exotropia causing psychological problems of patients and parents (see Appendix A for detail). The inclusion criteria were: male or female patients aged 7 to 17 years old with basic-type IXT who received strabismus surgery one month prior and had a successfully corrected alignment (≤10 PD exodeviation or ortho when fixating at distance and near targets using PACT). The exclusion criteria included those with esophoria or complaints of diplopia, best-corrected visual acuity (BCVA) less than 20/25, anisometropia >1.50 diopter in spherical equivalent refraction (SER), or those with dysfunction of oblique muscle or oculomotor incoordination. More detailed eligibility criteria are described in this article [25].

### 2.3. Enrolment and Randomization Grouping

The eligible participants were informed about the study and provided consent prior to participating. Patients were randomly assigned to the orthoptic therapy group or the control group using a random number generator from SPSS software version 20.0 (IBM Corp., Armonk, NY, USA) with equal probability. The flowchart is shown in Figure 1.

### 2.4. Evaluations and Intervention Methods

Each enrolled patient completed strabismic and binocular visual function examinations. The control of the exodeviation was assessed at distance (6 m) and near (1/3 m) using the Office Control Score [28], which ranges from 0 (phoria) to 5 (constant exotropia). Levels 5 to 3 mean that the patient still shows exotropia sometimes, while levels 2 to 0 indicate that the patient can be controlled in an exophoria state. So, we assessed levels 5 to 3 when the patient fixated at a distance target of 4 m for an initial 30-s period of observation and repeated at near fixation of 33 cm for another 30-s period. If the patient remained alignment, levels 2 to 0 were then graded as the worst of three rapidly sequential trials. First, we evaluated the distance fixation and then near fixation. Then, we placed an occluder over the right/left eye for 10 s and then removed, measuring the length of time it takes for fusion to become re-alignment when the patient fixated the distance target. The worse level of control observed was recorded following three 10-s periods of occlusion. Near stereopsis was measured using the TNO test (Laméris Ootech B.V., Nieuwegein, the Netherlands) at 40 cm, and distant stereopsis was measured using the Distance Randot Stereotest (DRS, Stereo Optical Co., Inc., Chicago, IL, USA) at 3 m. Sensory fusion was tested using four Worth dots at 40 cm and 5 m fixation. The amplitude of fusional convergence [19] was assessed using a 1 to 40 PD fixed horizontal prism bar, both at distance (3 m) and then near (1/3 m); ocular alignment was assessed using the SPCT and PACT at distance (6 m) and near (33 cm). For our present study of fusional convergence, there were four possible scenarios: (1) if patients were unable to fuse the visual targets, the break point was recorded as 0; (2) if patients were able to control or motor fuse the exodeviation in free space, we assumed that their convergence fusion was at least equal to the magnitude of the measured angle of deviation; (3) additional convergence fusion was measured by determining the fusion break point and establishing it as the convergence reserve; (4) the sum of the angle of deviation and the convergence reserve was established as the total convergence amplitude.

For patients in the control group, their follow-up time points were 6 and 12 months. If refractive status changed, an appropriate glasses prescription was provided. Patients in the orthoptic therapy group were referred to the orthoptic therapist and were asked to receive at least 8 weeks of hospital-based orthoptic therapy (1–2 times per week, 60 min each time) along with home reinforcement (15–30 min, 5 days per week). The therapy protocol in our present study is based on participants’ binocular vision status and was modified based on the Convergence Insufficiency Treatment Trial randomised clinical trial [29]. The treatment program had three phases. In each phase, there were a number of subcategories. Each procedure had a designated endpoint that was obtained before moving on to the next level or phase. The details and the endpoint of the orthoptic therapy schedule were described in our protocol. The entire procedure was administered appropriately by a proficient orthoptic therapist. They had close contact with patients by phone or on Wechat. If the patients had any questions about the training or eye alignment, they would give guidance and encourage motivation. If a decrease in the test results was observed during the follow-up, the appropriate therapeutic treatment would be reinitiated. In addition, the therapist would contact the patients by phone on a monthly basis to review the home therapy procedure, motivate the patients to adhere to training and estimated compliance. Patients who had completed ≥10 hospital-based therapy and ≥30 times (75% of the number of home training that should be completed) complete home training recorded by their parents were considered as excellent cooperation. Patients were considered as a dropout if they had met the following criteria: (1) patients in orthoptic therapy group who did not accept training within 2 weeks of enrollment, (2) patients in the control group who had started any kind of postoperative non-surgical treatment (e.g., patching, orthoptic training), and (3) patients who requested to withdraw from the study.

### 2.5. Follow-Up Visits

Baseline data were assessed on the day of enrolment. Subjects made visits at 6 months ± 2 weeks, which were the 6- and 12-month follow-up visits. At each follow-up visit, all procedures were performed by two clinicians who were naive to the type of clinical intervention. If one or more protocol-specified criteria for suboptimal surgical outcome were met during any follow-up examination, the test was repeated by another investigator after a 10-min break; suboptimal surgical outcome was declared only when both the test and repeat test results met the criteria. Investigators recorded the results on the case report forms and data were subsequently entered into the EpiData 3.1. system.

### 2.6. Safety

Since the orthoptic therapy that was used in our study was a form of functional training (i.e., a noninvasive operation), the risk of harm to patients was considered to be minimal. Despite this assumption, we nevertheless monitored for possible ocular and systemic adverse events throughout the study using emails or other types of social media platforms, including instant messaging services.

### 2.7. Statistical Analysis

The baseline distributions of the participants’ characteristics, such as age, height, weight, SER, and exo-deviation were expressed with means (standard deviation, SD) or medians (interquartile range, IQR), depending on whether the data were normally distributed or not. We checked for the normality of our datasets using a Shapiro–Wilk test and found that our data were normally distributed (p’s > 0.05). Categorical variables were expressed as the rates (or proportions).

We determined the necessary sample size based on previous studies [14,23] by performing a power analysis with 80% power and a type I error probability (alpha) of 0.05. Finally, we concluded that a sample size of 136 would be adequate to answer our main research questions.

We considered both motor alignment (the deviation between 10 PD of exophoria/exotropia to 5 PD of esophoria/esotropia by SPCT at distance or near) and sensory status (loss of 2 octaves or more of stereopsis from baseline) when defining the successful outcome criteria. For the primary outcome, we cited a recent PEDIG study which used the “suboptimal surgical outcome” as the primary outcome. The cumulative suboptimal surgical outcome proportion of patients by 12 months was compared between the two groups using the Kaplan–Meier method. An intergroup difference and a corresponding 95% confidence interval (CI) were also calculated. Suboptimal surgical outcomes were defined as: (1) exodeviation of ≥10 PD at distance or near using SPCT, (2) constant esotropia ≥6 PD at distance or near using SPCT, or (3) loss of 2 octaves or more of stereopsis from baseline [14].

Secondary outcomes were surgical motor alignment success, stereopsis, exodeviation speculated by PACT, fusional control score, and fusional convergence parameters. Surgery motor alignment success was defined as exotropia of <10 PD and esotropia of <5 PD [30] According to the results of DRS and TNO stereopsis, we divided the patients into three groups: “Good” denotes DRS ≤ 100″ or TNO ≤ 60″, “moderate” indicates 100″ < DRS ≤ 400″ or 60″ < TNO ≤ 480″, and “nil” indicates that they were unable to recognize the cues [31,32]. Stereopsis was transformed into log units for data analysis. Patients with “nil” stereopsis were assigned to the next highest 0.3 log increment level (i.e., 800 arcsec for DRS and 960 arcsec for TNO) [33]. A fusional exotropia control score of ≤2 indicates that the patient can be controlled in an exophoria state [28]. We divided the patients into two groups according to the exotropia control scores at the 12-month follow-up.

For all IXTs who completed the 12-month follow-up, we conducted a repeated-measures analysis of variance (ANOVA) to compare the difference in exodeviation and fusional convergence amplitude between the two groups. We used a two-tailed chi-square test to compare the difference in stereopsis grouping and exotropia control grouping between the orthoptic therapy and control groups. Statistical analysis was performed using SPSS software version 20.0 (IBM Corp., Armonk, NY, USA). A *p* value of 0.05 was established as statistically significant.

## 3. Results

### 3.1. Characteristic of Participants

Overall, 136 IXT patients were enrolled in our study. The mean age was 10.4 ± 2.5 years, and 70 (51.4%) were male. The mean preoperative exodeviation angle was −34.2 ± 9.6 PD at a distance and −38.6 ± 9.4 PD nearby. All of our patients obtained good alignment and binocular fusion function one month after surgery. Participant characteristics of each group are shown in Table 1. There were no significant differences in baseline demographic and clinical characteristics between these two groups.

Of the follow-up cohort, 129 patients (94.8%) and 117 patients (86.0%) across both groups returned for their 6-month and 12-month visits, respectively. No patients dropped out due to the side effect associated with orthoptic therapy. There were no significant differences between the baseline data for subjects who completed and did not complete the study (all ps > 0.05; Table 2).

### 3.2. Compliance of the Orthoptic Therapy Group

According to the hospital’s training records and home training logs, we found that 29 patients strictly followed the three phases’ training plan that is described in our protocol; 11 patients received at least 8 weeks hospital-based training but did not cooperate adequately with the family training; 19 patients received less than 10 times therapy in hospital, and there was no record of home training; another 9 withdrew from the study because participating in the study was too time-consuming for them.

### 3.3. Suboptimal Surgical Outcome

At the 6-month follow-up visit, suboptimal surgery outcomes occurred in 9 orthoptic receivers and 17 controls. At the 12-month follow-up time, the cumulative suboptimal surgery outcome occurred in 14 (9 at 6-month postoperative, added 5 at 12-month postoperative) orthoptic receivers and 29 (17 at 6-month postoperative, added 12 at 12-month postoperative) controls. The cumulative probability of suboptimal surgical outcome by 12 months was 20.5% (14/68) in the orthoptic therapy group and 42.6% (29/68) in the control group. There was a significant difference between these two groups (treatment group difference of orthoptic therapy group minus control group, −22.1%; 95% CI, −38.4% to −5.7%; χ^2^ = 7.40, *p* = 0.007; see Figure 2 and Table 3).

### 3.4. Surgical Motor Alignment Success Outcome

Surgical motor alignment success was defined as exotropia of <10 PD and esotropia of <5 PD. During this 12-month follow-up, 9 and 10 participants in the orthoptic therapy group and control group withdrew from the study, respectively. We excluded these 19 patients and compared the surgical motor alignment success outcome of the two groups at 12-months, finding that there was a significant difference between the orthoptic therapy group (83%, 49/59) and control group (60.3%, 35/58) (treatment group difference of orthoptic therapy group minus control group, −21.8%; 95% CI, −37.95% to −5.65%; χ^2^ = 6.327, *p* = 0.012).

### 3.5. Stereopsis

Based on the results of DRS and TNO stereopsis, we separately divided the patients into three subgroups: good, moderate and nil. According to the DRS and TNO 1-month postoperative results, there was no significant difference between the orthoptic therapy and control groups (χ^2^ = 0.12, *p* = 0.94; χ^2^ = 0.21, *p* = 0.34, respectively). In the case of the data from the 6-month postoperative visit, there was a significant difference in distant stereopsis between the two groups (χ^2^ = 9.55, *p* = 0.008) but not for near stereopsis (χ^2^ = 5.83, *p* = 0.054). We observed similar results for the 12-month postoperative outcomes (χ^2^ = 7.85, *p* = 0.02 for DRS and χ^2^ = 1.32, *p* = 0.249 for TNO; Table 4).

The mean improvement in distance stereopsis (log DRS) at 12 months was 0.3 ± 0.4 log seconds of arc for the orthoptic therapy group and 0.1 ± 0.4 log seconds of arc for the control group. There was a significant difference between the two groups (t = 2.9, *p* = 0.004). However, the improvement in near stereopsis (log TNO) between the two groups was not significantly different (0.2 ± 0.3 vs. 0.1 ± 0.3, t = 0.53, *p* = 0.597).

### 3.6. Exo-Deviation Drift

There was no significant difference in the amount of exodeviation between the two groups at baseline (at near: t = 0.67, *p* = 0.504; at distance: t = 0.13, *p* = 0.890). The comparison of three follow-up visits at 1, 6 and 12 months after the operation showed that patients in the two groups gradually drifted outward (at near: F = 38.91, *p* < 0.001; at distance: F = 33.77, *p* < 0.001). In the orthoptic therapy group and the control group, the mean exodeviation drift in PACT magnitude over 12 months was −3.9 ± 6.6 PD vs. −5.1 ± 6.2 PD at distance (t = 0.91, *p* = 0.364) and −3.8 ± 7.8 PD vs. −7.2 ± 7.9 PD at near (t = 2.26, *p* = 0.025). However, Figure 3 shows that there was no significant difference in exodeviation magnitude at near (F = 3.78, *p* = 0.055) or at distance (F = 0.92, *p* = 0.339) between the two groups during the three follow-up visits.

### 3.7. Fusional Exotropia Control

In the orthoptic therapy group and control group, the proportions of patients with distance exotropia control of 2 or better at the 12-month visit were 86.4% (51/59) and 67.2% (39/58), and those with near exotropia control were 91.5% (54/59) and 67.2% (39/58), respectively (see Table 5). There was a significant difference between the two groups (χ^2^ = 6.07, *p* = 0.016 for distance and χ^2^ = 12.2, *p* = 0.001 for near).

### 3.8. Fusional Convergence Amplitudes

The fusion image convergence amplitude was significantly improved from the 1- to 12-month follow-up visits in the orthoptic therapy group (at near: from 19.4 ± 12.1 PD to 30.2 ± 12.9 PD, F = 13.38, *p* < 0.001; at distance: from 13.3 ± 10.2 PD to 25.1 ± 13.5 PD, F = 21.39, *p* < 0.001). The most notable improvement was from 1 to 6 months post-surgery, which was maintained until the 12-month visit (see Figure 4). For the patients in the control group, there was no significant improvement during these three follow-up visits (at near: from 19.7 ± 13.4 PD to 21.1 ± 13.0 PD, F = 1.05, *p* = 0.352; at distance: from 13.2 ± 11.7 PD to 14.3 ± 12.1 PD, F = 0.06, *p* = 0.944, respectively) (see Figure 4). For 29 patients with excellent cooperation compared to the 30 patients with poor cooperation, it was found that there was a significant difference in fusional convergence amplitude between them (29.8 ± 11.9 PD vs. 20.0 ± 13.4 PD, Z = 195.5, *p* = 0.008 at distance and 34.3 ± 11.4 vs. 25.9 ± 13.2 PD, Z = 198.5, *p* = 0.004 at near, respectively).

A mixed ANOVA revealed that the total convergence amplitude was significantly different among time points of the visit (at near: F = 35.90, *p* < 0.001; at distance: F = 31.29, *p* < 0.001). In addition, there was a significant difference between the two groups (at near: F = 6.61, *p* = 0.012 at distance: F = 17.05, *p* < 0.001) and a significant interaction between the two variables (at near: F = 4.05, *p* = 0.021 at distance: F = 11.07, *p* < 0.001) (see Figure 5).

## 4. Discussion

Our prospective, randomised controlled trial demonstrates that postoperative orthoptic therapy can be effective in reducing the probability of suboptimal surgery outcomes for IXT children. In addition, we found a significant improvement in binocular function (e.g., stereopsis, fusional exotropia control, and fusional convergence reserve) in the orthoptic training group. However, we also found a persistent exodrift in both groups, although the amount of exodrift was smaller in the orthoptic therapy group.

The definition of successful outcome of surgery can vary across studies. At present, most researchers take motor status into account for a criterion of success. Kushner et al. considers esophoria/tropia ≤5 PD to exophoria/tropia ≤10 PD as a successful outcome after surgery [34]. Wu Haixiang has defined successful motor alignment as within 8 prism diopters (PD) (exo or eso) [35]. Jeoung defines a satisfactory outcome as within 10 PD of esotropia and exotropia during 6 months of follow-up after surgery [36]. In addition, some researchers have considered both motor and sensory status when defining the successful outcome criteria [14,37]. The primary outcome in our study was suboptimal successful outcome, defined as esotropia ≤5 PD to exotropia ≤10 PD tested by SPCT, combined with sensory status. There were two reasons for this: first, from motor alignment, a manifest tropia of specific magnitude could be more important than the total exodeviation for postoperative IXT; second, the main purpose of orthoptic therapy is to improve the fusional control and decrease the frequency and amount of manifest exotropia.

Success rates of surgery for IXT range from 43% to 85% [14,36,38,39], which is a large range. This discrepancy is due to different standards that have been used in defining and evaluating success, and incorporating different follow-up times for assessment. The success rate decreases with a longer follow-up time. For example, Lim et al. [12] reported the success rate of 58.1% after R&R (lateral rectus recession and medial rectus resection) at 1 year and they found that it decreased to 46.9% at 2 years. A recent PEDIG study reports that the suboptimal successful outcome of 27% at 1-year postoperative increases to 40% at 3 years [14]. While the suboptimal successful rate in our control group was 42.6% at 12-month follow-up, it is quite higher than that reported in previous studies. This difference could be due to our inclusion criteria, which were <10 PD exodeviation or ortho at 1 month postoperatively; patients with exophoria after surgery could experience exotropia recurrence over follow-up, with a higher probability. Surgery re-establishes the balance of power of the extraocular muscles. However, there are remodelling changes in both treated and untreated muscles [40,41] and adaptive responses within the neuron after patients undergo extraocular muscle surgery [42]. This may be related to the persistent exodrift and the high rate of under-correction after the surgery.

Orthoptic therapy has been used as a nonsurgical treatment to improve the control of deviation, thereby reducing the occurrence of exotropia. Some studies support the notion that orthoptic therapy should be combined with surgery, pre- or post-operatively or both [43,44]. However, the role of orthoptic training in the success outcome of surgery is unclear. Our results show that the surgical motor/sensory success rate can be significantly better in the orthoptic therapy group than in the control group. In addition, the distance stereopsis and fusional function parameters (e.g., fusional control, fusional convergence reserve and the total fusional convergence amplitude) significantly improved, suggesting the superiority of orthoptic therapy and agreeing with previous reports about its binocular benefits [44,45]. We assumed that better fusional function would contribute to a stable postoperative outcome. Further research is needed to explore the role of mechanisms likely associated with exodeviation control to identify the non-surgical therapeutic implications [46].

There are two limitations in the study. First, the postoperative follow-up time was relatively brief, lasting for only 12 months. While Yam et al. [21] found that better fusional convergence could predict the following 3 years of good control in 117 Chinese children with IXT, we assumed better post-operative functions (e.g., sensory fusion, motor fusion, proprioceptive awareness) could contribute to a stable surgical outcome. Furthermore, we will continue to observe these patients for longer follow-up, i.e., 24 months. Second, we only recruited patients with less than 10 PD exodeviation; the limitation of our recruitment does not clarify whether the effect of orthoptic therapy can be the same for individuals who have a larger exodeviation or esodeviation.

## 5. Conclusions

In conclusion, our randomised, controlled study confirmed that early postoperative orthoptic therapy combined with home reinforcement orthoptic therapy may decrease the incidence of suboptimal surgical outcomes and improve fusional control and binocular function in IXT. It did not improve it for all participants and there was still an increase in the size of the deviation postoperatively. These findings provide further evidence that surgery combined with orthoptic therapy is an alternative treatment option for treating IXT to those with good compliance.

## Figures and Tables

**Figure 1 jcm-12-01283-f001:**
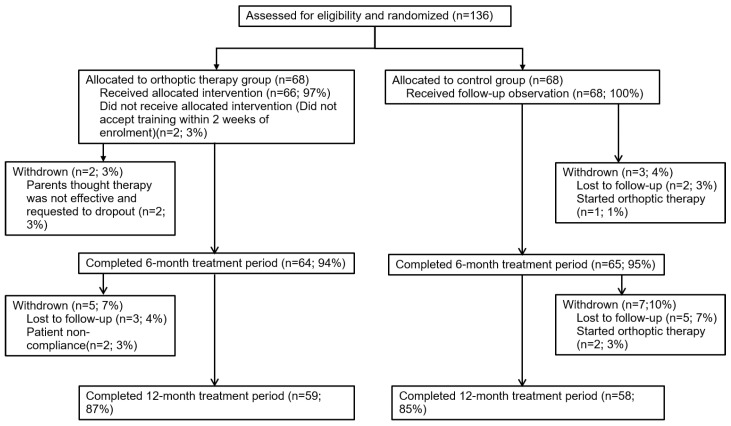
Flowchart showing the progress of participants through our current study.

**Figure 2 jcm-12-01283-f002:**
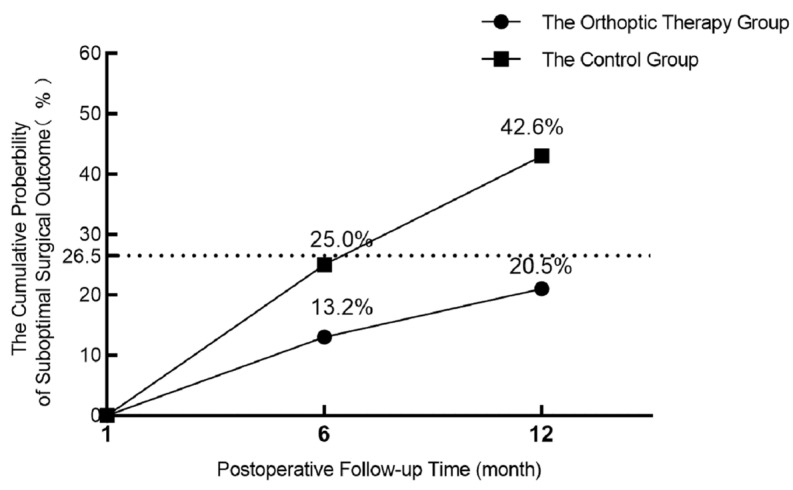
Graph showing the cumulative probability of suboptimal surgical outcome by 12 months from Kaplan–Meier analysis (n = 136). The dotted line represents the cumulative probability of suboptimal surgical outcomes at 12 months after surgery in intermittent exotropia patients, as shown in a previous article.

**Figure 3 jcm-12-01283-f003:**
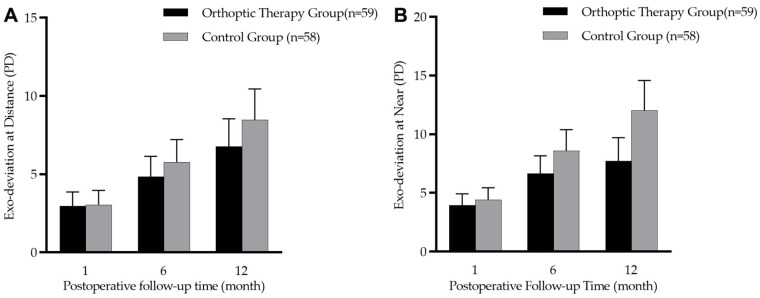
Box diagram showing postoperative 1-, 6-, and 12-month prism and alternate cover test (PACT) magnitudes in the orthoptic therapy group (n = 59) and the control group (n = 58). (**A**): distance deviation; (**B**): near deviation.

**Figure 4 jcm-12-01283-f004:**
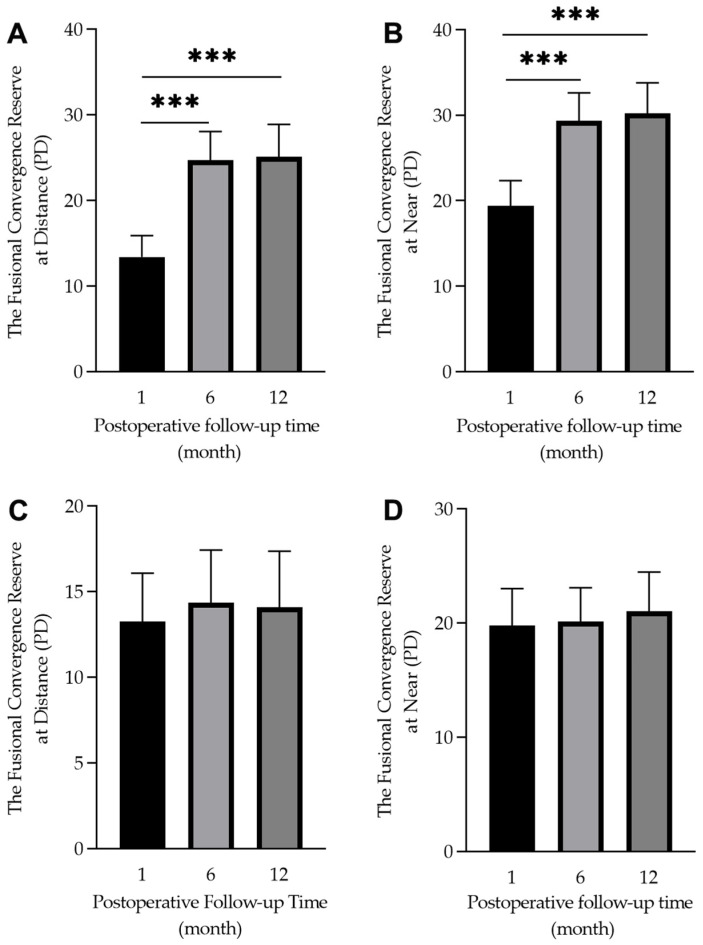
Box diagram showing the 1-, 6-, and 12-month postoperative fusional convergence reserve in the orthoptic therapy group (**A**,**B**) and the control group (**C**,**D**). There was a significant difference within the group (*** *p* < 0.001).

**Figure 5 jcm-12-01283-f005:**
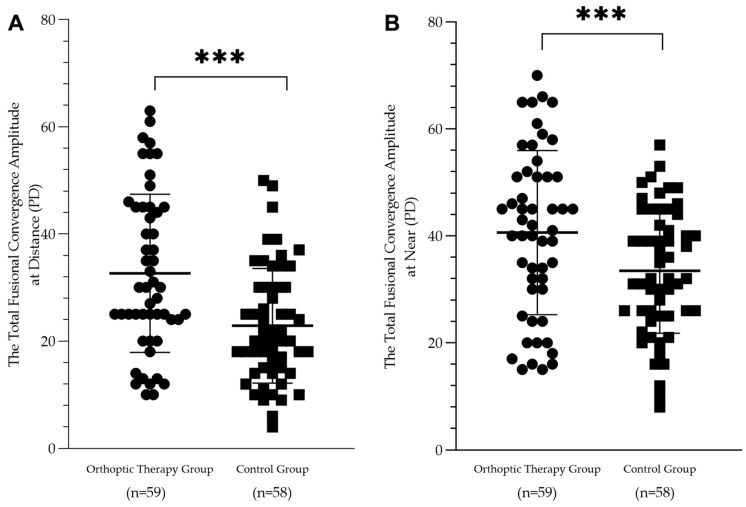
Scatterplots showing the postoperative 12-month total fusional convergence amplitude in the orthoptic therapy group and the control group at distance (**A**) and at near (**B**). There was a significant difference between the two groups (*** *p* < 0.001).

**Table 1 jcm-12-01283-t001:** Demographic and Baseline Clinical Characteristics of Two Groups.

Characteristic		OTG (n = 68)	CG (n = 68)	*p* Value
Preoperative				
Sex (F:M)		30:38	36:32	0.954
Age, y, mean (SD)		10.2 (2.6)	10.6 (2.4)	0.340
Height, cm, mean (SD)		142.6 ± 15.2	145.2 ± 13.7	0.298
Weight, Kg, mean (SD)		37.1 ± 14.0	38.6 ± 11.6	0.470
SER, D, mean (SD)	OD	−1.1 ± 1.3	−1.6 ± 1.6	0.036
	OS	−1.1 ± 1.4	−1.5 ± 1.5	0.089
Exo-deviation, PD, mean (SD)	Distance	−33.3 ± 10.9	−33.9 ± 9.9	0.738
	Near	−37.3 ± 10.5	−38.4 ± 10.2	0.554
R&R:BLRc:ULRc		53:14:1	43:22:3	0.148
Stereopsis (logarcsec)	Distance	2.7 ± 0.3	2.8 ± 0.2	0.200
	Near	2.4 ± 0.4	2.3 ± 0.4	0.159
Post-operative				
Fusional convergence reserve (PD)	Distance	13.4 ± 10.2	13.2 ± 11.7	0.942
	Near	19.4 ± 12.1	19.7 ± 13.4	0.878
Deviation (PD)	Distance	−2.9 ± 3.7	−3.1 ± 3.7	0.890
	Near	−3.9 ± 3.9	−4.4 ± 4.2	0.504
Stereopsis (logarcsec)	Distance	2.5 ± 0.4	2.5 ± 0.4	0.718
	Near	2.1 ± 0.3	2.2 ± 0.3	0.735

Means ± standard deviations for age, height, weight, SER, exo-deviation, stereopsis, fusional convergence reserve. OD, Oculus dexter (right eye); OS, Oculus sinister (left eye); SER, spherical equivalent of refraction; D, Diopter; PD, Prism Diopter; OTG: Orthoptic Therapy Group; CG: Control Group; R&R, unilateral recess-resect; BLRc, bilateral lateral rectus recession; ULRc, unilateral lateral recession.

**Table 2 jcm-12-01283-t002:** The difference in demographic and baseline clinical characteristics between the IXT patients who missed and completed the 12-month follow-up visit.

Characteristic		Completed (n = 117)	Missed (n = 19)	*p* Value
Preoperative				
Sex (F:M)		57:60	9:10	0.913
Age, y, mean (SD)		10.2 ± 2.5	11.3 ± 2.3	0.065
Height, cm, mean (SD)		142.6 ± 15.2	142.9 ± 14.3	0.048
Weight, Kg, mean (SD)		37.1 ± 14.0	37.5 ± 13.1	0.346
SER, D, mean (SD)	OD	−1.3 ± 1.5	−1.0 ± 1.1	0.461
	OS	−1.3 ± 1.5	−1.1 ± 1.3	0.567
Exo-deviation, PD, mean (SD)	Distance	−34.1 ± 9.6	−35.0 ± 9.7	0.602
	Near	−38.5 ± 9.4	−39.1 ± 9.9	0.645
Stereoacuity (logarcsec)	Distance	3.7 ± 0.6	3.8 ± 0.5	0.526
	Near	2.3 ± 0.4	2.3 ± 0.5	0.461
Post-operative				
Fusional convergence reserve (PD)	Distance	13.5 ± 11.1	11.6 ± 10.1	0.522
	Near	19.6 ± 12.7	19.4 ± 13.1	0.942
Deviation (PD)	Distance	−3.1 ± 3.7	−2.1 ± 3.3	0.271
	Near	−4.2 ± 4.1	−3.7 ± 3.7	0.618
Stereoacuity (logarcsec)	Distance	2.5 ± 0.4	2.5 ± 0.3	0.794
	Near	2.1 ± 0.3	2.2 ± 0.3	0.318

Means ± standard deviations for age, height, weight, SER, exo-deviation, stereoacuity, fusional convergence reserve. OD, Oculus dexter (right eye); OS, Oculus sinister (left eye); SER, spherical equivalent of refraction; D, Diopter; PD, Prism Diopter; OTG: Orthoptic Therapy Group; CG: Control Group.

**Table 3 jcm-12-01283-t003:** Suboptimal Surgical Outcome by 12 months in the Orthoptic Therapy Group and Control Group.

	6 Months		12 Months		Total	
	OTG	CG	OTG	CG	OTG	CG
No. at risk	68	68	59	51	N/A	N/A
No. with suboptimal surgical outcome	9	17	5	12	14	29
Constant XT	5	11	4	10	9	21
Stereo loss	3	4	1	2	4	6
Constant XT and Stereo loss	1	2	0	0	1	2
Cumulative probability of suboptimal surgical outcome (%)	13.2	25	20.5	42.6	N/A	N/A

OTG: Orthoptic Therapy Group; CG: Control Group; No.: number; Stereo loss: decrease in near or distant stereopsis 2 octaves or more from enrolment, or to nil, confirmed by a retest; XT: exotropia prism ≥10 PD diopters by simultaneous prism and cover test at distance or near, confirmed by a retest; N/A: not applicable.

**Table 4 jcm-12-01283-t004:** Distribution of stereopsis subgroup in the orthoptic therapy group and control group at different follow-up visits.

Parameters		The Orthoptic Therapy Group			The Control Group		
		1 Month (n = 68)	6 Months (n = 64)	12 Months (n = 59)	1 Month (n = 68)	6 Months (n = 65)	12 Months (n = 58)
DRS	Good	12	27	21	13	13	13
	Moderate	20	27	31	21	30	26
	Nil	36	10	7	34	22	19
TNO	Good	14	29	27	14	17	21
	Moderate	48	32	32	52	46	37
	Nil	6	3	0	2	2	0

**Table 5 jcm-12-01283-t005:** Fusional Exotropia Control Score by 12 months in the orthoptic therapy group and the control group.

	Distance		Near	
Fusional Exotropia Control Score	OTG n (%)	CG n (%)	OTG n (%)	CG n (%)
(0) No exotropia unless dissociated, recovers <1 s (phoria)	35 (59.3)	25 (43.1)	32 (54.2)	24 (41.4)
(1) No exotropia unless dissociated, recovers 1–5 s	12 (20.3)	12 (20.7)	17 (28.8)	14 (24.1)
(2) No exotropia unless dissociated, recovers >5 s	5 (8.5)	4 (6.9)	6 (10.2)	3 (5.2)
(3) Exotropia <50% of 30-s observation	5 (8.5)	4 (6.9)	3 (5.1)	12 (20.7)
(4) Exotropia >50% of 30-s observation	2 (3.4)	7 (12.1)	1 (1.7)	4 (6.9)
(5) Constant exotropia		6 (10.3)		1 (1.7)
Mean (SD)	0.7 (1.1)	1.5 (1.8)	0.7 (0.9)	1.3 (1.4)

OTG: Orthoptic Therapy Group; CG: Control Group; SD: standard deviations.

## Data Availability

Data from the study are available on reasonable request from the corresponding author.

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
