# Peer review of "The Effects of Orthoptic Therapy on the Surgical Outcome in Children with Intermittent Exotropia: Randomised Controlled Clinical Trial"

_jcm, 2023, doi:10.3390/jcm12041283_

Round 1

Reviewer 1 Report

This is a randomized study of patients following surgery for intermittent exotropia.  The design does not state what criteria were used to determine the need for surgery.  If control score changes were used, or parameters of loss of stereoacuity, these are not stated.  The postoperative result was the same in the 2 arms of the study though.  Patients were randomized to 8 weeks of orthoptic exercises described in another paper with follow up exercises at home.  The patients deviation, fusional amplitudes, stereoacuity at distance and near, and fusional reverses were measured at 6 months and 1 year after enrollment.  This study supports others that orthoptic therapy following surgery may improve fusional outcomes at 12 months.  There is still an increase in the size of the residual deviation of both groups over 12 months, but the control and fusional reserve are improved with orthoptic therapy.  We do not know the importance of improvement of those parameters in long-term control. 

Few other points:

Please give reference to current treatment success rates to put into context.  Also need rates of continued success at one year, or recurrence rates at one year.  Coffey B, Wick B, Cotter S, et al. Treatment options in intermittent exotropia:  a critical appraisal. Optom Vis Sci 1992;386-404.

In addition, need more info on orthoptic treatment in XT. --Ma, MM, Kang Y, Chen C, et al. Vision therapy for intermittent exotropia:  a case series. J Optom. 2021, 14:247-253.

Line 149 should be DRS (distance randot stereo), not RDS.  Continues thereafter in text as RDS.

The authors need to indicate how many participants continued with the home exercises 15 minutes daily 5X weekly during the 12 month period.  What was compliance with home therapy?  How did they record, ask.  Otherwise both groups are just having observation.

Line 321.  Change to:  Orthoptic therapy may decrease the incidence of suboptimal .  It did not improve it for all participants and there was still increase in the size of the deviation.

Otherwise interesting paper.

Reviewer 2 Report

In the manuscript, the authors analyzed the effect of orthoptic therapy on surgical outcome in children with Intermittent exotropia (X(T)).

The paper is clear and well-structured however many relevant revisions must be done.

lines 48-52: The authors describe only few studies with surgical outcome in X(T). All of them show very low percentage of surgical success without the postoperative help of orthoptic therapy. Most paper report higher success rates (as in ref 11 of the paper itself).

Lines 72-80. The authors should explain why they include only successfully corrected patients and not all the patients with X(T) surgically corrected. Do the authors want to demonstrate that the orthoptic postoperative therapy useful only in the successfully corrected patients? Moreover, some authors have suggested that successful alignment is more likely when there is a planned initial overcorrection of the deviation (Hardesty 1978; Keech 1990; Koo 2006; Raab 1969; Scott 1981), anticipating resolution of the overcorrection in the first six weeks following surgery (Mitchell 2000).

lines 109-110: the authors should describe the compliance of orthoptic therapy in the results

Lines 141: The authors should explain why they use the "suboptimal surgical outcome proportion" as primary outcome and not, more easily, the deviation between 10 Δ of exophoria or exotropia to 5 Δ of esophoria or esotropia by SPCT at distance or near, as widely used.

Lines 149: One year of follow-up is very short.

Lines 191-203: It is not clear how the authors calculate the cumulative percentage of suboptimal surgical outcome. As written in Table 3 it seems that they sum the data at 6 months with those at 12months. A patient with suboptimal surgical outcome at 6 months and at 12months is counted twice?

Lines 228-237: The authors should explain why they use the “exodeviation drift” in the text and for the data analysis and not in Figure 3, where the PACT magnitudes are described. The data shown in Figure 3 seem different from the data presented in Table 3. 

Table 3: It is written “ET” in the Table. Do the authors mean “XT”?

The paper includes an excessive number of self-citations.

The reference “Hwang JM. How to Better Treat Patients with Intermittent Exotropia: A Review of Surgical Treatment of Intermittent Exotropia. Korean J Ophthalmol. 2022 Oct” must be added to bibliography and commented by the authors

The reference   “Pang Y, Gnanaraj L, Gayleard J, Han G, Hatt SR. Interventions for intermittent exotropia. Cochrane Database Syst Rev. 2021 Sep 13;9(9)” must be added to bibliography and commented by the authors

Round 2

Reviewer 1 Report

This is a very interesting article about the effect of adding convergence exercises at near following surgery for basic intermittent exotropia.  The efficacy of adding exercises is compared to a control group.  Baseline characteristics are about the same in the two groups, but my largest concern is the number of patients assigned to therapy that did not complete therapy appropriately. Other minor comments.

Please clarify if the office based score was the average of 3 different assessments or just one assessment.  The score has both a distance and near assessment from 0-5.  How was the near control handled?

Did the enrollment criteria include only basic XT?  No patients with convergence insufficiency type?  What surgery was performed, bilateral lateral rectus recession or only R/R.

Line 173 enrollment- spelling error

Compliance with therapy is an issue.  Only 29 patients followed therapy guidelines.  Hard to understand why the fusional convergence remained so much higher in group that didn’t do any further vergence exercises following the 8 weeks of in hospital training.  Was there a difference in outcome sensory or motor of those who only did partial therapy? 

How many were compliant with home therapy?  How was this assessed?

Line 217:  verb should be followed instead of referred

Line 269:  Authors state that the fusional vergence was defined as the deviation plus further reserve where fusion still possible.  The average seems to imply that most patients couldn’t fuse their deviation?  The average deviation was 33 and the fusional convergence was 13 PD, so does that mean 43 PD of convergence.

Line 413 should be Kushner et. al.

Line 431:  Need to define R & R, lateral rectus recession and medial rectus resection.  Were plications allowed?

Did authors assess parent/child desire for further surgery at 12 months?  Since it was one of the criteria to proceed with surgery, it would be interesting to know if those that did therapy felt less like they needed to correct the deviation than the control group.

Line 469:  should say:   we assumed that better fusional function would contribute to a stable postoperative outcome.

Line 472:  exodeviation postoperatively (should be added)

Conclusion is too strong, given lack of compliance with orthoptic therapy.

Reviewer 2 Report

The paper was revised as requested

Author Response

There was no comment for Round 2. I really appreciate it.